# The Pandemic’s Impact on Mental Well-Being in Sweden: A Longitudinal Study on Life Dissatisfaction, Psychological Distress, and Worries

**DOI:** 10.3390/ijerph22060952

**Published:** 2025-06-17

**Authors:** Daniel Lindberg, Kent W. Nilsson, Jonas Stier, Birgitta Kerstis

**Affiliations:** 1Department of Social Work, School of Behavioural, Social and Legal Sciences, Örebro University, 70182 Örebro, Sweden; 2Center for Clinical Research, Central Hospital of Västerås, Uppsala University, 75310 Uppsala, Sweden; kent.nilsson@regionvastmanland.se; 3Division of Public Health Sciences, School of Health, Care and Social Welfare, Mälardalen University, 72134 Västerås, Sweden; 4Division of Social Work, School of Health, Care and Social Welfare, Mälardalen University, 72123 Västerås, Sweden; jonas.stier@mdu.se,; 5Division of Caring Sciences, School of Health, Care and Social Welfare, Mälardalen University, 72134 Västerås, Sweden; birgitta.kerstis@mdu.se

**Keywords:** COVID-19, life dissatisfaction, longitudinal study, mental health, psychological distress, sex differences, societal expectations, Sweden, worries

## Abstract

The COVID-19 pandemic has had a profound impact on society, including on physical and mental health. This study investigated changes in mental health parameters among a Swedish sample during and after the pandemic. Using a longitudinal study, we investigated the relationships among life dissatisfaction, psychological distress, and worries, with factors such as age, sex, education, severe illness, and job loss due to the pandemic among 588 men (mean age 54.9 years), and 653 women (mean age 52.9 years). The results reveal sex differences in life dissatisfaction and psychological distress; in addition, younger individuals reported more life dissatisfaction, and psychological distress compared with older groups. Women were 4.5 times more likely than men to report worries in 2021 and 2.3 times more likely in 2022, even after adjusting for age, education, severe illness, and job loss. This study shows how societal expectations and sex roles may exacerbate these differences in mental well-being during a time of crisis. The conclusions emphasize the importance of considering factors such as sex, age, education, and employment status when developing interventions and support systems during a global crisis. The COVID-19 pandemic will continue to affect society for a long time, indicating a need for ongoing research into population-level consequences.

## 1. Introduction

In March 2020, the World Health Organization (WHO) declared COVID-19 a global pandemic; by July 2024, over 7 million deaths had occurred [1]. The pandemic has disrupted societies worldwide, impacting physical and mental health [2,3]. It has led to financial uncertainty, inflation, job loss, and declines in living standards and well-being [4]. The virus was and is still today a direct health threat and induces anxiety about illness and healthcare collapse.

However, crises can also elicit collective responses that enhance life satisfaction and impart a sense of purpose [5]. Some individuals may experience growth and new perspectives post-crisis, boosting life satisfaction [6,7]. As the pandemic’s impact on mental health varied across different demographics globally, Sweden’s unique approach to managing the pandemic offers valuable insights [8]. Understanding how sex, age, and psychological factors such as life satisfaction, psychological distress, and worries relate to each other is crucial for addressing the pandemic’s diverse impacts. Sweden had one of Europe’s lowest excess mortality rates, but Sweden’s death toll exceeded other Nordic countries combined, demonstrating the regional limitations of the strategy [9]. Research indicates that individuals with moderate physical activity (PA) levels report greater life satisfaction and happiness than those with low PA levels [10]. Increased PA was also associated with higher life satisfaction during the pandemic [11]. A systematic review found that anxiety prevalence was lower among the general population than among COVID-19-infected individuals (27.3% vs. 39.6%) and higher in women compared with men (47.8% vs. 27.8%) [12]. The COVID-19 pandemic has significantly impacted on the mental health of the population, leading to profound effects on individual and collective well-being both during and likely after the crisis [13]. An Irish study describes that the prevalence of major depression and generalized anxiety disorder did not increase because of, or during the early phase of the COVID-19 pandemic [14].

Women reported higher levels of anxiety and distress during the pandemic compared with men [15,16]. Age also played a role, with older adults generally experiencing less pandemic-related stress and better psychological well-being than younger individuals [17]; this suggests that age may have mitigated some of the negative impacts of the pandemic. Life satisfaction was found to be linked to both age and sex, with younger adults often struggling more to maintain well-being during crises [18]. Younger adults also reported higher psychological distress than older adults, although this difference diminished over time, and prior diagnoses of anxiety or depression predicted greater distress [17]. A study from Turkey found that fear of COVID-19 correlated with lower life satisfaction and greater psychological distress [19]. In Germany, increased depressive symptoms, loneliness, and decreased life satisfaction were noted among vulnerable groups such as younger individuals and those with a history of mental disorders during a 12-month follow-up [20]. The COVID-19 pandemic is described in Slovakia, having lasting negative effects on the mental and physical health of young adults, significantly impacting their overall quality of life [21]. Sweden welfare system with its universalistic policies, relatively generous and equalizing social and economic benefits in combination with its comparatively different approach to the COVID-19 pandemic focusing on public compliance with recommendations and personal responsibility makes it an interesting case for scientific enquiry [22]. Hence, the present study investigated changes in mental health factors in a Swedish sample during and after COVID-19, focusing on life dissatisfaction, psychological distress, and worries, and examines their relationship with the demographics of age, sex, education, serious illness due to COVID-19, and job loss.

## 2. Materials and Methods

### 2.1. Recruitment of Participants and Data Collection

This longitudinal descriptive and comparative study is based on three web surveys from the “Values in a Crisis” World Values Survey [23]. This panel has been widely used in research and featured in studies published in international journals [24,25]. Participants were recruited through NOVUS and comprised Swedes aged 18–79. They received an information letter and a link to a 20-min online questionnaire; by completing the questionnaire, participants were considered to have consented to participate. Response rates were 56% for the first survey (S1) in 2020, 75% for the second survey (S2) in 2021, and 60% for the third survey (S3) in 2022. A total of 1241 participants completed all surveys. All personal data was deleted after collection according to the Helsinki Declaration [26] and Swedish law [27]. Prior studies using the same data have showed that the dropout only have minimal impact on the results during the COVID-19 pandemic [25] The surveys included a question about; How afraid are you that you or your loved ones get sick and suffer severely from the Corona virus? answered by 5 options.

### 2.2. Measures

#### 2.2.1. Life Dissatisfaction

Participants rated their life dissatisfaction during the pandemic using a Likert scale from Very satisfied (0) to Very dissatisfied (9) across five aspects: Health condition, Financial situation, Social relations, Work-life balance, and Life as a whole [28]. Cross-construct comparisons: The parallel scaling allows for direct comparison of effect sizes and correlations between different psychological constructs within the same study. Uniform scaling reduces the risk of reverse-coding mistakes during data analysis, which is a common source of analytical errors. Higher scores consistently indicate life dissatisfaction, making statistical relationships easier to interpret. An index was created, with higher values indicating life dissatisfaction. Cronbach’s alpha reliability coefficients were 0.848 for S1, 0.852 for S2, and 0.850 for S3.

#### 2.2.2. Psychological Distress

Psychological distress over the past 2 weeks was assessed using five items inspired by Löve [29]: (1) Feeling nervous, anxious, or on edge; (2) Inability to stop or control worrying; (3) Feeling down, depressed, or hopeless; (4) Little interest or pleasure in activities; and (5) Feeling lonely. Responses ranged from Not at all (0) to Nearly every day (3). An index was created, with higher scores indicating a greater level of psychological distress. Cronbach’s alpha was 0.860 for S1, 0.823 for S2, and 0.884 for S3.

#### 2.2.3. Worries Related to COVID-19

Participants were asked about their concerns regarding the current COVID-19 outbreak with specific focus on the following aspects: (1) contracting the virus, (2) unknowingly infecting others, (3) the necessity of permanent lifestyle changes, and (4) the adequacy of healthcare system resources. Response options ranged from Not at all worried (0) to Very worried (3), resulting in an overall range of 0–12. An index was created, with higher values indicating greater levels of concern. Cronbach’s alpha scores were 0.631 for S1, 0.644 for S2, and 0.706 for S3.

#### 2.2.4. Statistical Analysis

Descriptive statistics include frequencies, line graphics and percentages for categorical variables and means with standard deviations (SDs) for continuous variables. Participants were divided into four age groups: young adults (18–26, *n* = 121, 10%), adults (27–45, *n* = 360, 29%), middle-aged adults (46–64, *n* = 358, 29%), and older adults (65–80, *n* = 402, 32%). Education was categorized into Compulsory/senior high school and University. Kolmogorov–Smirnov test was used to assess normality. None of the instruments were normally distributed. The Mann-Whitney U test was used to compare sex and education levels with life dissatisfaction, psychological distress, and worries. The Spearman rho correlation was applied to assess relationships between the three instruments. The Wilcoxon signed rank test was adopted to analyze differences across three time points (1 vs. 2, 1 vs. 3, 2 vs. 3). The Kruskal-Wallis test was used to compare differences among the four age groups. Logistic regression analysis was used to test the relationships among life dissatisfaction, psychological distress, and worries, while controlling for age, education, loss of work, and sex. The results are presented as odds ratios (ORs) with 95% confidence intervals (CIs). The CI provides a range of values within which the true population parameter is likely to fall with 95% certainty. The association is considered statistically significant if the CI does not overlap 1.00. All tests were two-tailed with statistical significance at *p* ≤ 0.05. Analyses were performed using IBM SPSS Statistics (v. 29.0).

## 3. Results

### 3.1. Description of the Sample

The average age of the participants was 53.8 years (SD = 16.3), with men averaging 54.9 years (SD = 15.6) and women 52.9 years (SD = 16.7). There were 588 men (47.4%) and 653 women (52.6%). Regarding educational level, 56.4% had a university degree. Serious illnesses from COVID-19 infection significantly increased over time, at 3.5% in S1 (*n* = 43), 5.8% in S2 (*n* = 72), and 9.8% in S3 (*n* = 122). Few participants reported job losses due to the pandemic (S1: 3.0%, *n* = 37; S2: 3.4%, *n* = 42; S3: 2.2%, *n* = 27), with no significant changes observed over time. Sweden has one of the world’s highest average life expectancies, with 82.3 years for men and 85.4 years for women [30].

### 3.2. Sex, Age, and Education Differences

Life dissatisfaction showed no sex differences but increased from S1 to S2 for both men and women. Men experienced an increase from S1 to S3 (*p* = 0.019), while women saw a decline from S2 to S3 (*p* = 0.009). The youngest age group reported most life dissatisfaction, and the oldest the lowest. University graduates reported the lowest dissatisfaction at S2 (*p* = 0.041) than those with compulsory or senior high education. Both education groups saw an increase from S1 to S2, but participants with compulsory/senior high school reported decrease levels from S2 to S3.

Women consistently reported higher psychological distress (Q35–Q39) than men (*p* < 0.001, *p* < 0.001, *p* = 0.003) at all survey point times, with significant decreases in S3 for both sexes. The youngest age group had the highest psychological distress, particularly in S2, while the oldest had the least, noting reductions to S3. No educational differences were found in psychological distress, although both groups reported declines.

Women reported more worries than men at all survey time points (*p* < 0.001, *p* < 0.001, *p* < 0.001). Total score differences were noted across all surveys, with the biggest drop occurring in S3. The youngest age group had the most worries in S1 and S2; the oldest had fewer worries, with the largest decrease occurring between S2 and S3. Worries decreased across all ages from S1 to S3, except for the youngest group, which showed no significant difference between S1 and S2. Educational levels showed no differences, but both groups increased between S1 and S2 and decreased in S3 (see Figure 1, and Table 1).

A hierarchical regression showed that sex has a significant effect on worries (B = 0.021, *p* < 0.001), sex effect remains significant when controlling for education (B = 0.020, *p* < 0.001) and that sex effect persists when controlling for both education and age (B = 0.020, *p* < 0.001).

Parallel lines: No interaction—sex effect is consistent across education levels. Non-parallel lines: Significant interaction—sex effect varies by education level. *Y*-axis values: Based on the regression constant (≈0.95) and coefficients. Higher value: Indicate greater worry levels during the pandemic (Figure 2). Non-parallel lines in S1 and S2 showed a significant interaction in worries where sex effect varies by education level. The effect of being women on worries was stronger among those with lower education (B = 0.035) compared to higher education (B = 0.010). The parallel lines in S3 the sex difference in worries (B ≈ 0.020) remains consistent regardless of education level (Figure 2).

### 3.3. Correlations

The three scales consistently correlated across all survey points. Psychological distress and worries showed the strongest relationship, while life dissatisfaction and worries had a weaker but still significant correlation over time (see Table 2).

Age showed a moderate negative correlation with life dissatisfaction (S1: −0.217, S2: −0.223, S3: −0.244) and psychological distress (S1: −0.068, S2: −0.107, S3: −0.179), suggesting that older individuals were more satisfied and less distressed during the pandemic. Age did not significantly correlate with worries. Sex had no significant correlation with life dissatisfaction but exhibited moderate positive correlations with psychological distress during the pandemic (S1: 0.165, S2: 0.158) and a slight positive tendency post-pandemic (S3: 0.085). Women experienced greater psychological distress and worries than men. Education slightly negatively correlated with life dissatisfaction in S2, indicating that higher education levels protected against life dissatisfaction, but it did not correlate with distress or worries. Serious illness from COVID-19 was moderately positively correlated with both life dissatisfaction and psychological distress in S2 (.113) and S3 (.108) and slightly positively with worries during the pandemic (S1: 0.068, S2: 0.057). Loss of work due to the pandemic showed moderate positive correlations with life dissatisfaction (S1: 0.126, S2: 0.151) and psychological distress (S1: 0.126, S2: 0.151) and very weakly correlated with worries (S1: 0.060, S2: 0.064), indicating a minimal impact on worry levels (see Table 3).

### 3.4. Binary Logistic Regression

Age consistently influenced life dissatisfaction, with younger individuals reporting more life dissatisfaction than older ones. During the pandemic (S1 and S2), sex affected life dissatisfaction, more so in S2. After the pandemic (S3), age and job loss slightly but significantly impacted life dissatisfaction, with younger people who lost jobs reporting lower satisfaction (see Table 4).

Psychological distress decreased with age, at OR 0.87 (CI 0.78–0.98). Women were 1.8 times more likely than men to experience higher psychological distress during the pandemic (S1 = 1.9, S2 = 1.8) and 1.3 times more likely after the pandemic (S3), adjusted for age, education, serious illness, and loss of work. Serious illness increased the likelihood of distress by 2.1 to 1.8 times during severe cases of COVID-19. Loss of work raised the risk of distress by 5.2 times in S1 (coefficient 5.162), although this effect was not significant in S2 and S3. In S2 and S3, predictors of psychological distress were serious illness, sex (women), and age (younger groups) (see Table 5).

There was no significant difference in worries in S1. In S2, women were 4.5 times more likely than men to report worries, at OR 4.5 (CI 1.0–21.2). The CI indicated variability in the results, suggesting that further research or a larger sample size could help narrow the CI and provide a more precise estimate. After adjusting for age, education level, serious illness, and loss of work, women were 2.3 times more likely than men to report worries after the pandemic, at OR 2.3 (CI 1.1–4.6) (see Table 6).

## 4. Discussion

This study investigated relationships among life dissatisfaction, psychological distress, worries, age, sex, education, COVID-19 illness, and job loss during and after the pandemic, highlighting the psychological and social impacts of COVID-19 on a Swedish population. Women reported more life dissatisfaction than men, likely due to increased caregiving responsibilities and workload during the pandemic, as well as societal expectations and sex roles. Effective support systems are needed to address these sex differences to meet women’s needs during crises [31]. Age played a significant role in life dissatisfaction, particularly among the youngest age group (aged 18–29), who consistently reported increase. This demographic faced numerous disruptions, including interruptions to education, delays in career progression, and increased social isolation, all of which may have contributed to more life dissatisfaction [32]. The uncertainty of their prospects during the pandemic added to their stress, highlighting the need for targeted interventions to support young adults during such times.

In contrast, individuals aged 50 and above exhibited more varied responses, with some improvements noted in S2, possibly due to a greater sense of resilience and adaptive coping strategies developed over time. A longitudinal study reported that depressive symptoms in older adults increased from 11.4% before the pandemic to 27.2% during the pandemic and remained above pre-pandemic levels (14.9%) after the pandemic [33]. Older adults also reported lower levels of stress and better psychological well-being [17].

Education influenced life dissatisfaction as well, according to a study that described those with lower educational attainment experiencing greater fluctuations compared with higher-educated individuals, who maintained more stable levels over time [34]. Higher education may provide better access to resources, more stable employment opportunities, and greater social support networks, contributing to more consistent levels of life dissatisfaction.

Additionally, job loss had a significant impact on life dissatisfaction, which was particularly noticeable in S3 after the pandemic. The economic uncertainties and financial stress associated with job loss likely exacerbated feelings of insecurity and dissatisfaction, underscoring the need for targeted support systems for those who became unemployed due to the pandemic. This finding aligns with a study indicating that work situation is associated with life dissatisfaction [35]. It is essential to develop employment policies and programs that can offer stability and support to individuals during economic downturns. However, a potential limitation is the low frequency (2–3%) of respondents reporting job loss, which may impact the stability and reliability of the regression results. While job loss showed a significant association with increased suffering and life dissatisfaction, it is important to interpret these findings with caution due to the low prevalence of job loss in our sample. We conducted additional robustness checks, including bootstrapping, which confirmed the direction of the effect but highlighted wider confidence intervals, underscoring the need for cautious interpretation.

Psychological distress was another critical area of concern, with women reporting more distress throughout the pandemic, especially during its initial phases. This trend is consistent with previous research suggesting that women are more likely to experience anxiety and distress [15]. The necessity for mental health resources and support for women during such times cannot be overstated, as women’s well-being significantly affects their families and communities. Interestingly, individuals aged 50 and above showed significant reductions in psychological distress over time—a trend also observed among those aged 18–29, although this group’s improvements were noted after the pandemic. The initial high levels of distress among younger age groups could be attributed to the uncertainty and disruption in these individuals’ personal and professional lives, while the subsequent decrease may indicate adaptation and the development of coping mechanisms [36,37]. This highlights the importance of fostering resilience and providing mental health support to all age groups during crises.

Both individuals with secondary education and those with higher education reported decreases in psychological distress, but those with lower educational attainment experienced most significant reduction over time. This finding suggests that, while education level can buffer against psychological distress, other factors such as social support, access to mental health resources, and economic stability also play crucial roles in mitigating distress. This finding emphasizes the need for comprehensive mental health strategies that consider educational and socioeconomic factors.

This study found that women and younger adults experienced more psychological distress and worries during the pandemic, which aligns with prior research on unemployment and health risks. Women reported more life dissatisfaction and higher psychological distress than men, often due to disproportionate domestic responsibilities [38]. Middle-aged women—particularly single parents or employed women—often support both children and elderly parents, offering emotional closeness, practical help, and financial assistance [39].

Older adults, especially those over 65 with comorbidities such as heart disease, diabetes, obesity, and respiratory conditions, faced increased vulnerability to severe illness and death from COVID-19 [40]. A meta-analysis confirmed higher mortality and hospitalization risks for older populations [41]. According to the Organisation for Economic Co-operation and Development [42], younger workers, particularly those under 30, faced disproportionately high rates of unemployment and underemployment during the pandemic due to the economic shutdown. Social distancing measures, lockdowns, and concerns about vulnerability to COVID-19 caused many older adults to experience increased feelings of loneliness, anxiety, and depression. Older adults, particularly those living alone, were at a heightened risk of mental health problems due to isolation during COVID-19 [43,44].

Younger people also faced considerable social and psychological strain during the pandemic. For example, disruptions to education, job insecurity, and social isolation affected their mental health [45]. A study indicated that rates of anxiety, depression, and substance use were higher among young adults during the pandemic, due in part to social relational stressors, economic stress, and uncertain futures [46]. A correlation was found in the cited study between the instruments, but this correlation was not particularly high.

However, these results need to be interpreted in relation to Sweden’s approach to the pandemic, emphasizing voluntary guidelines over strict mandates, highlights an intriguing balance between personal responsibility and public health measures. This strategy contrasts sharply with the more stringent policies adopted by countries like Italy and France during the same period [33]. This may have mitigated severe distress and worries during the pandemic.

In early 2021, Sweden was among the first eight EU countries to vaccinate 80% of individuals aged 80 and older with their first dose. However, disparities were noted; lower coverage was observed among home care recipients, high-risk groups, and specific healthcare personnel. Overall, vaccination rates declined between doses 2 and 3. Coverage was influenced by age, income, education, and birthplace, with Sweden-born individuals demonstrating the highest rates regardless of these factors.

Regardless of age, coverage is highest among people born in Sweden. It increases with age regardless of country of birth, and there is also a correlation between income level and coverage, regardless of country of birth [47]. Furthermore, the Swedish welfare system with its universalistic policies, relatively generous and equalizing social and economic benefits may have protected against even more gender and age inequalities during the pandemic [48].

This analysis highlights that mental health was influenced by sex and age during and after the pandemic, with women reporting more life dissatisfaction, psychological distress, and worries compared with men. Concerning age, younger individuals reported more life dissatisfaction, psychological distress and worries than older groups. These findings emphasize the need for targeted interventions to address psychological and social impacts of COVID-19 and future pandemic.

### Limitations and Strengths

This study’s strengths include its longitudinal design and large sample size, which offer insights into Sweden’s COVID-19 consequences. Despite its statistical power, limitations exist: Except for worries, the variables did not directly address COVID-19; the participants’ pre-pandemic status was unknown; high dropout rates might have skewed results; and 55% of the respondents had university education which is a significantly higher proportion than that in the general population of Sweden. While the data was sufficient for an analysis, caution is needed regarding generalization. A strength was that we were able to examine changes and recovery in life dissatisfaction, psychological distress and worries over time and during a global health crisis. As a result, we gain a deeper understanding of the most negatively affected groups in the sample during a global crisis and how different factors, particularly sex and age was associated with changes and recovery over time.

Future research should examine the long-term impacts of the pandemic to inform policies for well-being and resilience. Investigating factors such as social isolation, health risks, economic instability, sex roles, education, and employment can provide a comprehensive understanding and identify areas for targeted interventions.

## 5. Conclusions

The Swedish strategy during the pandemic might have contributed to the low proportion of people reporting life dissatisfaction during and after the pandemic. However, those who perceived a deterioration in life satisfaction, psychological distress, and worries may require further attention from the Swedish Welfare Authorities.

Our findings highlighted significant sex differences, with women reporting more life dissatisfaction, psychological distress and worries, suggesting the need for gender-sensitive public health interventions. Recovery patterns showed that while short-term distress eased, long-term impacts on life dissatisfaction endured, emphasizing the importance of sustained mental health support. Younger adults (18–29) faced the greatest life dissatisfaction, while older adults (65–79) exhibited remarkable resilience.

These insights underline the need for tailored policy responses sensitive to gender and age vulnerabilities, integrating psychosocial dimensions into broader strategies to promote recovery, resilience, and well-being across diverse demographic groups. Future research should continue to explore these dynamics to better understand long-term consequences of the pandemic and to inform policies and practices aimed at enhancing well-being and resilience in the face of such global crises.

## Figures and Tables

**Figure 1 ijerph-22-00952-f001:**
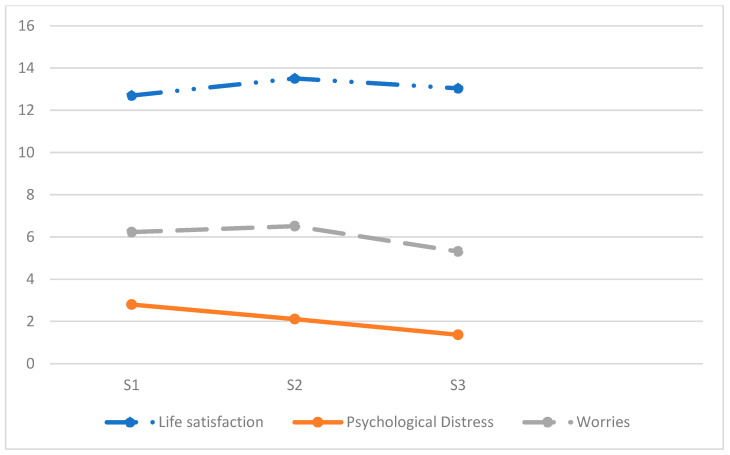
Life dissatisfaction, psychological distress and worries in the three timepoint.

**Figure 2 ijerph-22-00952-f002:**
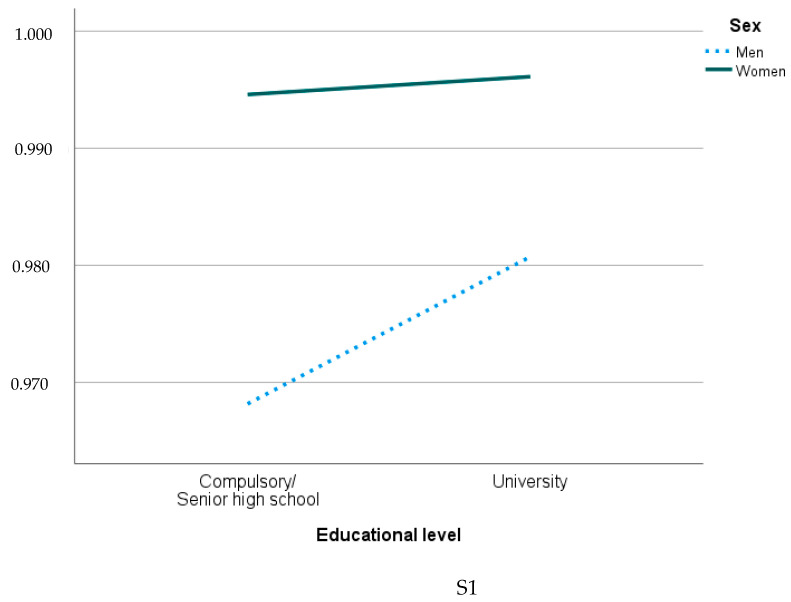
Worries in relation to sex and education in S1 (2020), S2 (2021) and S3 (2022).

**Table 1 ijerph-22-00952-t001:** (**a**) Life dissatisfaction, psychological distress and worries in the three timepoints (2020, 2021, and 2022) in relation to sex, age, and education. (**b**) Psychological distress in the three timepoints (2020, 2021, and 2022) in relation to sex, age, and education. (**c**) Worries in the three timepoints (2020, 2021, and 2022) in relation to sex, age, and education.

(a)							
	S1		S2	S3	*p*-Value Between S1 and S2	*p*-Value Between S1 and S3	*p*-Value Between S2 and S3
**Life dissatisfaction**							
Men *n* = 588	12.19 (7.74)		12.89 (8.12)	12.67 (7.78)	**0.001** ^a^	**0.019** ^a^	0.502 ^a^
Women *n* = 653	13.17 (8.52)		14.07 (8.88)	13.38 (8.45)	**0.001** ^a^	0.165 ^a^	**0.009** ^a^
*p*-value		0.132 ^b^	0.064 ^b^	0.259 ^b^			
Total *N* = 1241	12.70 (8.17)		13.51 (8.54)	13.04 (8.14)	**0.001** ^a^	**0.010** ^a^	**0.015** ^a^
**Age groups**							
18–29 years *n* = 121 (9.8%)	14.32 (8.33)		15.49 (8.18)	15.49 (8.38)	0.057 ^a^	**0.010** ^a^	0.865 ^a^
30–49 years *n* = 360 (29.0%)	14.17 (8.12)		14.94 (8.29)	14.58 (7.93)	**0.028** ^a^	0.165 ^a^	0.308 ^a^
50–64 years *n* = 358 (28.9%)	13.46 (8.09)		14.38 (9.05)	13.49 (7.98)	**0.001** ^a^	0.547 ^a^	**0.050** ^a^
65–79 years *n* = 402 (32.4%)	10.23 (7.67)		10.87 (7.79)	10.53 (7.80)	**0.008** ^a^	0.255 ^a^	0.119 ^a^
*p*-value total	**0.001** ^b^		**0.001** ^b^	**0.001** ^b^			
**Education**							
Compulsory/ Senior high school *n* = 540 (43.6%)	13.11 (8.14)		14.04 (8.78)	13.54 (8.37)	**0.001** ^a^	0.059 ^a^	**0.019** ^a^
University *n* = 698 (56.4%)	12.35 (8.15)		13.07 (8.30)	13.62 (7.88)	**0.002** ^a^	0.085 ^a^	0.214 ^a^
*p*-value	0.064 ^b^		**0.041** ^b^	0.074 ^b^			
(**b**)							
**Psychological Distress**							
Men *n* = 588	1.63 (2.60)		1.66 (2.51)	1.11 (2.17)	0.417 ^a^	**0.001** ^a^	**0.001** ^a^
Women *n* = 653	2.48 (3.18)		2.52 (3.22)	1.61 (2.81))	0.762 ^a^	**0.001** ^a^	**0.001** ^a^
*p*-value	**0.001** ^b^		**0.001** ^b^	**0.003** ^b^			
Total *N* = 1241	2.08 (2.95)		2.11 (2.94)	1.37 (2.54)	0.461 ^a^	**0.001** ^a^	**0.001** ^a^
**Age groups**							
18–29 years *n* = 121	2.88 (3.37)		3.31 (3.46)	2.36 (2.48)	0.098 ^a^	0.177 ^a^	**0.002** ^a^
30–49 years *n* = 360	2.08 (2.92)		2.18 (2.99)	1.56 (2.77)	0.461 ^a^	**0.001** ^a^	**0.001** ^a^
50–64 years *n* = 358	1.96 (2.72)		1.96 (2.94)	1.20 (2.42)	0.817 ^a^	**0.001** ^a^	**0.001** ^a^
65–79 years *n* = 402	1.94 (2.30)		1.83 (2.64)	1.07 (2.35)	0.709 ^a^	**0.001** ^a^	**0.001** ^a^
*p*-value	**0.004** ^c^		**0.001** ^c^	**0.001** ^c^			
**Education**							
Compulsory/Senior high school	2.13 (3.01)		2.08 (3.11)	1.34 (2.49)	0.597 ^a^	**0.001** ^a^	**0.001** ^a^
University	2.04 (2.84)		2.14 (2.80)	1.38 (2.54)	0.140 ^a^	**0.001** ^a^	**0.001** ^a^
*p*-value	0.733 ^b^		0.105 ^b^	0.629 ^b^			
(**c**)							
**Worries**							
Men *n* = 575	5.75 (2.12)		6.07 (2.31)	4.89 (2.19)	**0.001** ^a^	**0.001** ^a^	**0.001** ^a^
Women *n* = 653	6.67 (2.17)		6.91 (2.18)	5.69 (2.27)	**0.001** ^a^	**0.001** ^a^	**0.001** ^a^
*p*-value	**0.001** ^b^		**0.001** ^b^	**0.001** ^b^			
Total *N* = 1 241	6.23 (2.19)		6.51 (2.28)	5.31 (2.27)	**0.001** ^a^	**0.001** ^a^	**0.001** ^a^
**Age groups**							
18–29 years *n* = 121	6.42 (1.96)		6.66 (2.01)	5.47 (2.26)	0.076 ^a^	**0.001** ^a^	**0.001** ^a^
30–49 years *n* = 360	6.17 (2.09)		6.43 (2.06)	5.36 (2.26)	**0.001** ^a^	**0.001** ^a^	**0.001** ^a^
50–64 years *n* = 358	6.39 (2.21)		6.63 (2.33)	5.40 (2.36)	**0.012** ^a^	**0.001** ^a^	**0.001** ^a^
65–79 years *n* = 402	6.10 (2.31)		6.42 (2.45)	5.11 (2.21)	**0.001** ^a^	**0.001** ^a^	**0.001** ^a^
*p*-value	0.148 ^c^		0.475 ^c^	0.266 ^c^			
**Education**							
Compulsory/Senior high school *n* = 540	6.30 (2.27)		6.52 (2.45)	5.32 (2.22)	**0.001** ^a^	**0.001** ^a^	**0.001** ^a^
University *n* = 698	6.19 (2.13)		6.49 (2.14)	5.29 (5.31)	**0.001** ^a^	**0.001** ^a^	**0.001** ^a^
*p*-value	0.437 ^b^		0.953 ^b^	0.734 ^b^			

S1 = survey 1, 2020, S2 = survey 2, 2021, S3 = survey 3, 2022. ^a^ Wilcoxon signed rank test, ^b^ Mann-Whitney U Test, ^c^ Kruskal-Wallis test, bold = statistically significant.

**Table 2 ijerph-22-00952-t002:** Correlations between life dissatisfaction, psychological distress and worries in the three surveys calculated with Spearman rho.

	Survey 1 Correlation *p*-Value	Survey 2 Correlation *p*-Value	Survey 3 Correlation *p*-Value
Life dissatisfaction and Psychological distress	0.423 **	0.481 **	0.476 **
Life dissatisfaction and Worries	0.210 **	0.232 **	0.252 **
Psychological distress and Worries	0.486 **	0.396 **	0.335 **

S1 = survey 1, 2020, S2 = survey 2, 2021, S3 = survey 3, 2022. ** *p*-value < 0.001.

**Table 3 ijerph-22-00952-t003:** Correlation between age, sex, education, serious illness, loss of work, life dissatisfaction, distress, worries, and survey timepoints, calculated with Spearman’s *rho*.

	S1 Correlation *p*-Value	S2 Correlation *p*-Value	S3 Correlation *p*-Value
Age × Life dissatisfaction	−0.217 **	−0.223 **	−0.244 **
Sex × Life dissatisfaction	NS	NS	NS
Education × Life dissatisfaction	NS	−0.058 *	NS
Serious illness × Life dissatisfaction	NS	0.113 **	0.108 **
Loss of work × Life dissatisfaction	0.126 **	0.151 **	NS
Age × Psychological distress	−0.083 **	−0.107 **	−0.179 **
Sex × Psychological distress	0.165 **	0.158 **	0.085 **
Education × Psychological distress	NS	NS	NS
Serious illness × Psychological distress	NS	0.113 **	0.108 **
Loss of work × Psychological distress	0.126 **	0.151 **	NS
Age × Worries	NS	NS	NS
Sex × Worries	0.209 **	0.182 **	0.171 **
Education × Worries	NS	NS	NS
Serious illness × Worries	0.068 *	0.057 *	NS
Loss of work × Worries	0.060 *	0.064 *	NS

S1 = survey 1, 2020, S2 = survey 2, 2021, S3 = survey 3, 2022. NS = Not significant, * *p* < 0.05, ** *p* < 0.001.

**Table 4 ijerph-22-00952-t004:** Binary logistic regression for life dissatisfaction, survey timepoints 1, 2 and 3.

	Life Dissatisfaction S1		Life Dissatisfaction S2		Life Dissatisfaction S3	
	Exp (B) 95% Cl ^1^	Exp (B) 95% Cl ^2^	Exp (B) 95% Cl ^1^	Exp (B) 95% Cl ^2^	Exp (B) 95% Cl ^1^	Exp (B) 95% Cl ^2^
Age	**0.65 (0.50–0.84)**	**2.21 (1.34–3.64)**	**0.65 (0.48–0.88)**	**0.68 (0.50–0.93)**	**0.58 (0.42–0.80)**	**0.57 (0.42–0.78)**
Sex	**2.23 (1.36–3.67)**	**0.67 (0.52–0.88)**	**1.82 (1.03–3.22)**	**1.77 (0.10–3.14)**	1.07 (0.63–1.84)	1.04 (0.60–1.80)
Education	0.84 (0.52–1.37)	0.80 (0.49–1.30)	1.11 (0.64–1.94)	1.034 (0.59–1.82)	0.93 (0.54–1.60)	0.90 (0.52–1.58)
Serious illness	2.69 (0.36–19.79)	2.52 (0.39–18.81)	-	-	2.98 (0.72–12.40)	2.97 (0.70–12.54)
Loss of work	-	-	-	-	0.36 (0.10–1.23)	**0.22 (0.06–0.80)**

S1 = survey 1, 2020, S2 = survey 2, 2021, S3 = survey 3, 2022, bold = statistically significant. ^1^ Crude, ^2^ Adjusted for variables listed to the left. - = not possible.

**Table 5 ijerph-22-00952-t005:** Binary logistic regression for psychological distress survey timepoints 1, 2 and 3.

	Psychological Distress S1		Psychological Distress S2		Psychological Distress S3	
	Exp (B) 95% CI ^1^	Exp (B) 95% CI ^2^	Exp (B) 95% CI ^1^	Exp (B) 95% CI ^2^	Exp (B) 95% CI ^1^	Exp (B) 95% CI ^2^
Age	**0.84 (0.75–0.94)**	**0.87 (0.78–0.98)**	**0.82 (0.73–0.92)**	**0.84 (0.74–0.94)**	**0.68 (0.60–0.76)**	**0.68 (0.60–0.77)**
Sex	**1.88 (1.50–2.36)**	**1.86 (1.48–2.34)**	**1.82 (1.45–2.28)**	**1.78 (1.41–2.24)**	**1.36 (1.08–1.71)**	**1.32 (1.04–1.67)**
Education	1.13 (0.90–1.42)	1.092 (0.87–1.38)	1.18 (0.94–1.48)	1.13 (0.90–1.43)	1.07 (0.85–1.35)	1.05 (0.82–1.33)
Serious illness	**2.13 (1.08–4.18)**	**2.15 (1.08–4.27)**	**1.95 (1.16–3.29)**	**1.96 (1.15–3.32)**	**1.88 (1.29–2.73)**	**1.80 (1.23–2.64)**
Loss of work	**5.35 (2.07–13.82)**	**5.16 (1.98–13.47)**	1.76 (0.91–3.42)	1.45 (0.74–2.86)	1.32 (0.62–2.86)	0.95 (0.43–2.11)

S1 = survey 1, 2020, S2 = survey 2, 2021, S3 = survey 3, 2022, bold = statistically significant. ^1^ Crude, ^2^ Adjusted for variables listed to the left. CI = Confidence Interval.

**Table 6 ijerph-22-00952-t006:** Binary logistic regression for worries survey timepoint 1, 2 and 3.

	Worries S1		Worries S2		Worries S3	
	Exp (B) 95% CI ^1^	Exp (B) 95% CI ^2^	Exp (B) 95% CI ^1^	Exp (B) 95% CI ^2^	Exp (B) 95% CI ^1^	Exp (B) 95% CI ^2^
Age	0.60 (0.32–1.10)	0.60 (0.32–1.12)	1.27 (0.68–2.36)	1.36 (0.73–2.52)	0.94 (0.68–1.32)	0.97 (0.69–1.36)
Sex	2.81 (0.88–9.00)	2.63 (0.82–8.46)	4.49 (0.95–21.23)	4.40 (0.92–20.93)	**2.37 (1.18–4.77)**	**2.26 (1.12–4.55**)
Education	1.30 (0.45–3.72)	1,26 (0.44–3.65)	3.04 (0.78–11.82)	2.72 (0.70–10.61)	**1.93 (1.00–3.76)**	1.88 (0.96–3.69)
Serious illness	0.46 (0.06–3.60)	0.38 (0.05–3.09)	-	-	0.69 (0.24–1.80)	0.62 (0.24–1.64)
Loss of work	-	-	-	-	-	-

S1 = survey 1, 2020, S2 = survey 2, 2021, S3 = survey 3, 2022, bold = statistically significant. ^1^ Crude, ^2^ Adjusted for variables listed to the left. - = not possible. CI = Confidence Interval.

## Data Availability

Kerstis, Birgitta; Lindberg, Daniel (2025). FIG_share_250613.sav.figshare. Dataset. https://doi.org/10.6084/m9.figshare.29312996.v1.

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
