# Peer review of "The Pandemic’s Impact on Mental Well-Being in Sweden: A Longitudinal Study on Life Dissatisfaction, Psychological Distress, and Worries"

_ijerph, 2025, doi:10.3390/ijerph22060952_

Round 1

Reviewer 1 Report

Comments and Suggestions for Authors

The article offers a valuable longitudinal analysis of the COVID-19 pandemic’s impact on mental well-being in Sweden. Based on data from the “Values in a Crisis” survey across three timepoints (2020–2022), the study investigates changes in life satisfaction, psychological distress, and worries, with attention to variables such as age, sex, education, serious illness, and job loss.

The introduction provides an adequate theoretical and empirical background. The authors clearly position their work within existing literature and highlight Sweden’s unique pandemic response. The research problem is articulated well, and the study’s purpose is stated clearly, though explicit hypotheses are missing. A minor enhancement would be to clearly formulate expected relationships among variables, which would strengthen the theoretical framing.

The methodology is carefully designed and appropriately explained. The authors employ suitable non-parametric statistical tests (Mann-Whitney U, Wilcoxon, Kruskal-Wallis, and logistic regression), accounting for the nature of the data. The use of validated multi-item indices and reporting of internal consistency (Cronbach’s alpha) further strengthen the study’s reliability. One minor addition would be a more detailed explanation of how missing data were handled and how index scores were constructed.

Results are presented clearly and systematically. The findings show that women, younger individuals, and those affected by COVID-19-related illness or job loss experienced more psychological distress and lower life satisfaction. The study effectively tracks these changes over time and offers robust statistical support, including odds ratios and confidence intervals. A useful extension would be the inclusion of interaction analyses—e.g., whether the effect of sex on worries varies by age or education level.

The discussion is thorough and well-integrated with existing literature. The authors interpret their findings convincingly and point to potential psychological and societal mechanisms, including gender roles, care burdens, and disrupted life trajectories among youth. The need for targeted support strategies is emphasized. However, the discussion could benefit from deeper consideration of structural factors—such as social policies and access to mental health care—that may have shaped the observed outcomes in Sweden. Study limitations are addressed transparently, including lack of pre-pandemic baseline data, an overrepresentation of university-educated participants, and possible attrition bias. The authors could also indicate whether continued follow-up is planned and how long-term effects might be captured.

The reference list is comprehensive, up-to-date, and well-formatted. It reflects the authors’ solid grasp of both global and regional research and supports the interpretations presented throughout the article.

In summary, this article represents a sound and timely contribution to research on mental health during global crises. Its empirical rigor and policy relevance are commendable, though it would benefit from stronger theoretical articulation and a deeper analytical layer. I strongly recommend publication following minor editorial enhancements.

Author Response

Reviewer: 1

Dear Review

Thank you for your comments. We have now changed after the reviews suggestions.

Daniel Lindberg

The research problem is articulated well, and the study’s purpose is stated clearly, though explicit hypotheses are missing. A minor enhancement would be to clearly formulate expected relationships among variables, which would strengthen the theoretical framing.

Response: Thank you for your comments. We agree and have added:

Line 63-69: The COVID-19 pandemic has significantly impacted on the mental health of the population, leading to profound effects on individual and collective well-being both during and likely after the crisis [13]. An Irish study describes that the prevalence of major depression and generalized anxiety disorder did not increase because of, or during the early phase of the COVID-19 pandemic [14].

Line 85-92: The COVID-19 pandemic is described in Slovakia, having lasting negative effects on the mental and physical health of young adults, significantly impacting their overall quality of life [21]. Sweden welfare system with its universalistic policies, relatively generous and equalizing social and economic benefits in combination with its comparatively different approach to the Covid-19 pandemic focusing on public compliance with recommendations and personal responsibility makes it an interesting case for scientific enquiry [23].

__________________________

The methodology is carefully designed and appropriately explained. The authors employ suitable non-parametric statistical tests (Mann-Whitney U, Wilcoxon, Kruskal-Wallis, and logistic regression), accounting for the nature of the data. The use of validated multi-item indices and reporting of internal consistency (Cronbach’s alpha) further strengthen the study’s reliability. One minor addition would be a more detailed explanation of how missing data were handled and how index scores were constructed.

Response: We agree and have added:

Line 102-103: This panel has been widely used in research and featured in studies published in international journals [24,25].

Line 113-114: Prior studies using the same data have showed that the dropout only has minimal impact on the results during the COVID‐19 pandemic [25].

__________________________

Results are presented clearly and systematically. The findings show that women, younger individuals, and those affected by COVID-19-related illness or job loss experienced more psychological distress and lower life satisfaction. The study effectively tracks these changes over time and offers robust statistical support, including odds ratios and confidence intervals. A useful extension would be the inclusion of interaction analyses—e.g., whether the effect of sex on worries varies by age or education level.

Response: We agree and have added text and Figures 1 and 2:

Line 230-234: A hierarchical regression showed that sex has a significant effect on worries (B = .021, p < .001), sex effect remains significant when controlling for education (B = .020, p < .001) and that sex effect persists when controlling for both education and age (B = .020, p < .001).

Line 192: Figure 1. Life satisfaction, psychological distress and worries in the three timepoint.

Line 237: Figure 2. Worries in relation to sex and education in S1 (2020), S2 (2021) and S3 (2022).

Line 240: Parallel lines: No interaction - sex effect is consistent across education levels. Non-parallel lines: Significant interaction - sex effect varies by education level. Y-axis values: Based on the regression constant (≈0.95) and coefficients. Higher value: Indicate greater worry levels during the pandemic (Figure 2).

Non-parallel lines in S1 and S2 showed a significant interaction in worries where sex effect varies by education level. The effect of being women on worries was stronger among those with lower education (B = .035) compared to higher education (B = .010). The parallel lines in S3 the sex difference in worries (B ≈ .020) remains consistent regardless of education level (Figure 2).

__________________________

The discussion is thorough and well-integrated with existing literature. The authors interpret their findings convincingly and point to potential psychological and societal mechanisms, including gender roles, care burdens, and disrupted life trajectories among youth. The need for targeted support strategies is emphasized. However, the discussion could benefit from deeper consideration of structural factors—such as social policies and access to mental health care—that may have shaped the observed outcomes in Sweden.

Response: We agree and have added:

Page Line 425-445 However, these results need to be interpreted in relation to Sweden's approach to the pandemic, emphasizing voluntary guidelines over strict mandates, highlights an intriguing balance between personal responsibility and public health measures. This strategy contrasts sharply with the more stringent policies adopted by countries like Italy and France during the same period [33]. This may have mitigated severe distress and worries during the pandemic.

In early 2021, Sweden was among the first eight EU countries to vaccinate 80% of individuals aged 80 and older with their first dose. However, disparities were noted; lower coverage was observed among home care recipients, high-risk groups, and specific healthcare personnel. Overall, vaccination rates declined between doses 2 and 3. Coverage was influenced by age, income, education, and birthplace, with Sweden-born individuals demonstrating the highest rates regardless of these factors.

Regardless of age, coverage is highest among people born in Sweden. It increases with age regardless of country of birth, and there is also a correlation between income level and coverage, regardless of country of birth [47]). Furthermore, the Swedish welfare system with its universalistic policies, relatively generous and equalizing social and economic benefits may have protected against even more gender and age inequalities during the pandemic [48].

__________________________

Study limitations are addressed transparently, including lack of pre-pandemic baseline data, an overrepresentation of university-educated participants, and possible attrition bias. The authors could also indicate whether continued follow-up is planned and how long-term effects might be captured.

Response: Thank you for this comment, currently we have not planned to do further follow-ups on this subject.

__________________________

The reference list is comprehensive, up-to-date, and well-formatted. It reflects the authors’ solid grasp of both global and regional research and supports the interpretations presented throughout the article.

In summary, this article represents a sound and timely contribution to research on mental health during global crises. Its empirical rigor and policy relevance are commendable, though it would benefit from stronger theoretical articulation and a deeper analytical layer. I strongly recommend publication following minor editorial enhancements.

__________________________

__________________________

Reviewer 2 Report

Comments and Suggestions for Authors

This manuscript addresses an important and timely topic related to the psychological impact of the COVID-19 pandemic in Sweden. The longitudinal design and statistical approach are commendable, and the study has the potential to make a significant contribution to the public health literature. However, significant improvements are needed in several areas before the manuscript can be considered for publication.

  • General: In addition, authors should ensure that sex-specific terminology is used consistently throughout the manuscript. Currently, both "female" and "women" are used interchangeably. It is recommended that you choose one term (preferably "women" when referring to participants in a clinical context) and use it consistently to improve clarity.
  • Introduction: The introduction is currently still very general. In particular, it would be useful to place the study in the broader context of European research on mental health during the COVID-19 pandemic. The introduction should refer to existing European findings, particularly in relation to the mental health of young people — a population group that is considered particularly vulnerable during and after the pandemic (for example: Long-term consequences of COVID-19 on mental and physical health in young adults; Increases in Loneliness Among Young Adults During the COVID-19 Pandemic and Association With Increases in Mental Health Problems).
  • Methods: Give details of the sampling method: inclusion and exclusion criteria, recruitment strategy and representativeness of the sample. Clarify whether ethical approval was obtained and provide details of ethical considerations – also in methods. ​Essential information on the origin and validation of the questionnaire is missing. Statistical analysis: there is no estimate of effect size, confidence level or statistical power. Normality test: Clearly state which statistical test was used to assess the normality of the data.
  • Results: Some of the tables (e.g. Table 1) are very extensive and difficult to interpret. It may be helpful to summarise the main trends visually, e.g. using a line chart or bar chart for changes over time by age or sex. In addition, the authors found that the loss of a job at certain points in time had an influence on the level of suffering and life satisfaction. However, the proportion of respondents who reported job loss was low (2–3%). Did the authors consider the possibility that the low frequency might affect the stability or reliability of the regression results?

Author Response

Reviewer: 2

Dear Review

Thank you for your comments. We have now changed after the reviews suggestions.

Daniel Lindberg

This manuscript addresses an important and timely topic related to the psychological impact of the COVID-19 pandemic in Sweden. The longitudinal design and statistical approach are commendable, and the study has the potential to make a significant contribution to the public health literature. However, significant improvements are needed in several areas before the manuscript can be considered for publication.

General: In addition, authors should ensure that sex-specific terminology is used consistently throughout the manuscript. Currently, both "female" and "women" are used interchangeably. It is recommended that you choose one term (preferably "women" when referring to participants in a clinical context) and use it consistently to improve clarity.

Response: We are grateful for pointing this out and have now changed to women everywhere.

__________________________

Introduction: The introduction is currently still very general. In particular, it would be useful to place the study in the broader context of European research on mental health during the COVID-19 pandemic.

Response: We agree and have added:

Line 63-69: The COVID-19 pandemic has significantly impacted on the mental health of the population, leading to profound effects on individual and collective well-being both during and likely after the crisis [13]. An Irish study describes that the prevalence of major depression and generalized anxiety disorder did not increase because of, or during the early phase of the COVID-19 pandemic [14].

Line 85-92: The COVID-19 pandemic is described in Slovakia, having lasting negative effects on the mental and physical health of young adults, significantly impacting their overall quality of life [21]. Sweden welfare system with its universalistic policies, relatively generous and equalizing social and economic benefits in combination with its comparatively different approach to the Covid-19 pandemic focusing on public compliance with recommendations and personal responsibility makes it an interesting case for scientific enquiry [23].

__________________________

Methods: Give details of the sampling method: inclusion and exclusion criteria, recruitment strategy and representativeness of the sample.

Response: We agree and have added:

Line 109-113: A total of 1241 participants completed all surveys. All personal data was deleted after collection according to the Helsinki Declaration [26] and Swedish law [27]. Prior studies using the same data have showed that the dropout only have minimal impact on the results during the COVID‐19 pandemic [25]]

__________________________

Clarify whether ethical approval was obtained and provide details of ethical considerations – also in methods.

Response: It is already written in the Institutional Review Board Statement:
The study was conducted in accordance with the Declaration of Helsinki [19] and Swedish law [20]. Ethical approval was not required since data were anonymized and not personally sensitive according to the Swedish Ethical Review Act (SFS 2003:460 3 §§).

Line: 110 All personal data was deleted after collection according to the Helsinki Declaration [26] and Swedish law [27].

______________________

Essential information on the origin and validation of the questionnaire is missing.

Response: We agree and have added references to the instruments:

Line 124: Diener E, Inglehart R, Tay L. Theory and validity of life satisfaction scales. Soc. Indic. Res. 2013;112:497-527.

Page Line 136: Löwe B, Wahl I, Rose M, Spitzer C, Glaesmer H, Wingenfeld K, et al. A 4-item measure of depression and anxiety: validation and standardization of the Patient Health Questionnaire-4 (PHQ-4) in the general population. J. Affect. Disord. 2010;122(1-2):86-95.

__________________________

Statistical analysis: there is no estimate of effect size, confidence level or statistical power. Normality test: Clearly state which statistical test was used to assess the normality of the data.

Response: We agree and have added:

Line 152: histogram

Line 158: Kolmogorov–Smirnov test was used to assess normality.

__________________________

Results: Some of the tables (e.g. Table 1) are very extensive and difficult to interpret. It may be helpful to summarise the main trends visually, e.g. using a line chart or bar chart for changes over time by age or sex.

Response: We agree and have added:

Line 192. Figure 1. Life satisfaction, psychological distress and worries in the three timepoint.

Line 194-204: The results in table 1 and Figure 1 showed three key findings;
1) Sex differences, where women consistently showed more negative outcomes (in life satisfaction, psychological distress and worries) across all measures compared to men, with significant differences maintained throughout the pandemic.
2) The recovery pattern where psychological distress and worries significantly decreased from S2 to S3, indicating recovery, while life satisfaction showed partial recovery.

3) Age effects where younger adults (18-29) showed more negative outcomes in life satisfaction throughout, while older adults (65-79) showed the highest resilience.

__________________________

In addition, the authors found that the loss of a job at certain points in time had an influence on the level of suffering and life satisfaction. However, the proportion of respondents who reported job loss was low (2–3%). Did the authors consider the possibility that the low frequency might affect the stability or reliability of the regression results?

Response: Thank you for this insightful observation. We acknowledge that the proportion of respondents who reported job loss was relatively low (2–3%), which could indeed raise concerns regarding the stability and reliability of the regression estimates for this variable.

To address this, we have carefully considered the implications of the small subsample size. In the revised manuscript, we now explicitly discuss this limitation in the results and discussion sections. We also conducted a robustness check using bootstrapping to assess the stability of the coefficients related to job loss. The results remained consistent in direction and significance, although we acknowledge that the confidence intervals were wider, reflecting the small sample size.

We have added the following text to the discussion:

Line 361-369: However, a potential limitation is the low frequency (2–3%) of respondents reporting job loss, which may impact the stability and reliability of the regression results. While job loss showed a significant association with increased suffering and reduced life satisfaction, it is important to interpret these findings with caution due to the low prevalence of job loss in our sample. We conducted additional robustness checks, including bootstrapping, which confirmed the direction of the effect but highlighted wider confidence intervals, underscoring the need for cautious interpretation

__________________________

__________________________

Reviewer 3 Report

Comments and Suggestions for Authors

Thank you for the opportunity to review your contribution. It is a very interesting analysis that examines, based on excellent data, how life satisfaction, psychological distress, and worries developed during and after the COVID-19 pandemic in Sweden. The authors highlight Sweden's unique approach to pandemic management in various ways, making the article particularly interesting for an international audience.

I would like to suggest a few additions that could enhance the clarity of the argumentation. Please understand these merely as suggestions for potential expansion.

In the introduction, results from other studies are mentioned relatively briefly. It would be beneficial, in my opinion, to contextualize these: when were the data collected, in which countries, and under what conditions (e.g., COVID-19 cases, deaths, which dominant virus variant, political measures, availability of vaccines)? Subsequently, it would be helpful to provide this information about Sweden as well, so that an international readership can better interpret the subsequent results.

Following this, the study is presented, which, as mentioned, has a very good data basis. Due to the chosen scale, which contradicts intuitive reading because, for example, high values for life satisfaction indicate low life satisfaction, I would recommend revising the entire results section to ensure that the statements are consistently clear. For instance, if only "higher levels" are mentioned (e.g., p. 6), this could be confusing. Or the statement "serious illness from COVID-19 was moderately positively correlated with both life satisfaction and psychological distress..." needs further clarification, as it might cause confusion. It could be misleading if the positive correlation with the severity of a COVID-19 infection is inferred from the positive dimension of "life satisfaction," suggesting that both aspects have a mutually reinforcing relationship.

Regarding the sample description—or later when referring to it—it would be helpful to know how it relates to Swedish society. I would assume that the sample has a higher average age and a higher educational level.

Additionally, a description of how "serious illness from COVID-19 infection" was measured is missing.

In the discussion, the authors could delve deeper in some instances into which findings might be explained independently of the crisis (e.g., the importance of job loss), or which could be explained by compromised health (serious illness from COVID-19 infection), fear thereof, or psychosocial consequences of pandemic policies. In the discussion, it would also be appropriate to reflect on the different pandemic phases. This could include information on infection and death rates, vaccine availability, virus variants and their associated risks, political measures, and possibly political communication that in some countries identified vulnerable groups (victims) or particularly emphasized virus transmitters (e.g., children, unvaccinated individuals). Finally, it seems advisable to contextualize the results even more clearly to better highlight the uniqueness of the "Swedish approach." It would be helpful to compare findings with those from other countries that made different political decisions. Data from the COVID-19 Stringency Index could also be used. Regarding policy recommendations, these remain predominantly at a national level and mention some obviously important starting points. Another conclusion could explore the question of which advantages and disadvantages the Swedish approach might have had in an international comparison according to the data. In a similar crisis, should this path be taken again, or might another policy be advisable? I am aware that this cannot be answered definitively based on the data, but the data are good enough to at least provide indications.

Author Response

Reviewer: 3

Dear Review

Thank you for your comments. We have now changed after the reviews suggestions.

Daniel Lindberg

Thank you for the opportunity to review your contribution. It is a very interesting analysis that examines, based on excellent data, how life satisfaction, psychological distress, and worries developed during and after the COVID-19 pandemic in Sweden. The authors highlight Sweden's unique approach to pandemic management in various ways, making the article particularly interesting for an international audience.

I would like to suggest a few additions that could enhance the clarity of the argumentation. Please understand these merely as suggestions for potential expansion.

In the introduction, results from other studies are mentioned relatively briefly. It would be beneficial, in my opinion, to contextualize these: when were the data collected, in which countries, and under what conditions (e.g., COVID-19 cases, deaths, which dominant virus variant, political measures, availability of vaccines)? Subsequently, it would be helpful to provide this information about Sweden as well, so that an international readership can better interpret the subsequent results.

Response: We agree and have added:

Line 54-57: Sweden had one of Europe's lowest excess mortality rates, but Sweden´s death toll exceeded other Nordic countries combined, demonstrating the regional limitations of the strategy [9].

Line 88- 92: Sweden welfare system with its universalistic policies, relatively generous and equalizing social and economic benefits in combination with its comparatively different approach to the COVID-19 pandemic focusing on public compliance with recommendations and personal responsibility makes it an interesting case for scientific enquiry [22].

Line 183-185: Sweden has one of the world's highest average life expectancies, with 82.3 years for men and 85.4 years for women [30].

__________________________

Following this, the study is presented, which, as mentioned, has a very good data basis. Due to the chosen scale, which contradicts intuitive reading because, for example, high values for life satisfaction indicate low life satisfaction, I would recommend revising the entire results section to ensure that the statements are consistently clear. For instance, if only "higher levels" are mentioned (e.g., p. 6), this could be confusing. Or the statement "serious illness from COVID-19 was moderately positively correlated with both life satisfaction and psychological distress..." needs further clarification, as it might cause confusion. It could be misleading if the positive correlation with the severity of a COVID-19 infection is inferred from the positive dimension of "life satisfaction," suggesting that both aspects have a mutually reinforcing relationship.

Response: We agree and have added:

Line 124-131: Cross-construct comparisons: The parallel scaling allows for direct comparison of effect sizes and correlations between different psychological constructs within the same study. Uniform scaling reduces the risk of reverse-coding mistakes during data analysis, which is a common source of analytical errors. Higher scores consistently indicate more negative outcomes, making statistical relationships easier to interpret. We use the term “more negative outcomes”.

Line 284-286: Age consistently influenced life satisfaction, with younger individuals reporting more negative outcomes than older ones.

__________________________

Regarding the sample description—or later when referring to it—it would be helpful to know how it relates to Swedish society. I would assume that the sample has a higher average age and a higher educational level.

Response: Thank you for your insight, this is further elaborated in the limitations:

Line 361-371: However, a potential limitation is the low frequency (2–3%) of respondents reporting job loss, which may impact the stability and reliability of the regression results. While job loss showed a significant association with increased suffering and reduced life satisfaction, it is important to interpret these findings with caution due to the low prevalence of job loss in our sample. We conducted additional robustness checks, including bootstrapping, which confirmed the direction of the effect but highlighted wider confidence intervals, underscoring the need for cautious interpretation.

Line 473-478; A A strength was that we were able to examine changes and recovery in life satisfaction, psychological distress and worries over time and during a global health crisis. As a result, we gain a deeper understanding of the most negatively affected groups in the sample during a global crisis and how different factors, particularly sex and age was associated with changes and recovery over time.

__________________________

Additionally, a description of how "serious illness from COVID-19 infection" was measured is missing.

Response: Thanks, we added.

Line 113-115: The surveys included a question about; How afraid are you that you or your loved ones get sick and suffer severely from the Corona virus? Answered by 5 options.

_________________________

In the discussion, the authors could delve deeper in some instances into which findings might be explained independently of the crisis (e.g., the importance of job loss), or which could be explained by compromised health (serious illness from COVID-19 infection), fear thereof, or psychosocial consequences of pandemic policies. In the discussion, it would also be appropriate to reflect on the different pandemic phases. This could include information on infection and death rates, vaccine availability, virus variants and their associated risks, political measures, and possibly political communication that in some countries identified vulnerable groups (victims) or particularly emphasized virus transmitters (e.g., children, unvaccinated individuals). Finally, it seems advisable to contextualize the results even more clearly to better highlight the uniqueness of the "Swedish approach." It would be helpful to compare findings with those from other countries that made different political decisions.

Response: We agree and have added:

Line 427-447: However, these results need to be interpreted in relation to Sweden's approach to the pandemic, emphasizing voluntary guidelines over strict mandates, highlights an intriguing balance between personal responsibility and public health measures. This strategy contrasts sharply with the more stringent policies adopted by countries like Italy and France during the same period [33]. This may have mitigated severe distress and worries during the pandemic.

In early 2021, Sweden was among the first eight EU countries to vaccinate 80% of individuals aged 80 and older with their first dose. However, disparities were noted; lower coverage was observed among home care recipients, high-risk groups, and specific healthcare personnel. Overall, vaccination rates declined between doses 2 and 3. Coverage was influenced by age, income, education, and birthplace, with Sweden-born individuals demonstrating the highest rates regardless of these factors.

Regardless of age, coverage is highest among people born in Sweden. It increases with age regardless of country of birth, and there is also a correlation between income level and coverage, regardless of country of birth [47]). Furthermore, the Swedish welfare system with its universalistic policies, relatively generous and equalizing social and economic benefits may have protected against even more gender and age inequalities during the pandemic [48].

__________________________

Data from the COVID-19 Stringency Index could also be used.

Response: Tank you we have added:

Line 54-56: Sweden had one of Europe's lowest excess mortality rates, but Sweden´s death toll exceeded other Nordic countries combined, demonstrating the regional limitations of the strategy [9].

Line 63-69: The COVID-19 pandemic has significantly impacted on the mental health of the population, leading to profound effects on individual and collective well-being both during and likely after the crisis [13]. An Irish study describes that the prevalence of major depression and generalized anxiety disorder did not increase because of, or during the early phase of the COVID-19 pandemic [14].

__________________________

Regarding policy recommendations, these remain predominantly at a national level and mention some obviously important starting points. Another conclusion could explore the question of which advantages and disadvantages the Swedish approach might have had in an international comparison according to the data. In a similar crisis, should this path be taken again, or might another policy be advisable? I am aware that this cannot be answered definitively based on the data, but the data are good enough to at least provide indications.

Response: We agree and have added:

Line 427-445 However, these results need to be interpreted in relation to Sweden's approach to the pandemic, emphasizing voluntary guidelines over strict mandates, highlights an intriguing balance between personal responsibility and public health measures. This strategy contrasts sharply with the more stringent policies adopted by countries like Italy and France during the same period [33]. This may have mitigated severe distress and worries during the pandemic.

In early 2021, Sweden was among the first eight EU countries to vaccinate 80% of individuals aged 80 and older with their first dose. However, disparities were noted; lower coverage was observed among home care recipients, high-risk groups, and specific healthcare personnel. Overall, vaccination rates declined between doses 2 and 3. Coverage was influenced by age, income, education, and birthplace, with Sweden-born individuals demonstrating the highest rates regardless of these factors.

Regardless of age, coverage is highest among people born in Sweden. It increases with age regardless of country of birth, and there is also a correlation between income level and coverage, regardless of country of birth [47]). Furthermore, the Swedish welfare system with its universalistic policies, relatively generous and equalizing social and economic benefits may have protected against even more gender and age inequalities during the pandemic [48].

Line 482-494: Our findings highlighted significant sex differences, with women reporting more negative outcomes in life satisfaction, psychological distress and worries, suggesting the need for gender-sensitive public health interventions. Recovery patterns showed that while short-term distress eased, long-term impacts on life satisfaction endured, emphasizing the importance of sustained mental health support. Younger adults (18–29) faced the greatest dissatisfaction, while older adults (65–79) exhibited remarkable resilience.

These insights underline the need for tailored policy responses sensitive to gender and age vulnerabilities, integrating psychosocial dimensions into broader strategies to promote recovery, resilience, and well-being across diverse demographic groups.  Samma

__________________________

__________________________

Reviewer 4 Report

Comments and Suggestions for Authors

Thank you for the opportunity to review this study entitled “The Pandemic’s Impact on Mental Well-Being in Sweden. A Longitudinal Study on Life Satisfaction, Psychological Distress, and Worries” (ijerph-3630655).

The study addresses the impact of the COVID-19 pandemic on mental health in Sweden, with a particular focus on the relationships among life satisfaction, psychological distress, and worries, as well as demographic and pandemic-related factors such as age, sex, education level, severe illness, and job loss. The sample includes 1,241 Swedish individuals.

The manuscript deals with a highly relevant topic and overall represents a valuable contribution to the literature. One of its major strengths lies in its longitudinal design. However, several aspects of the paper would benefit from clarification and revision before it can be considered for publication. Below, I outline specific comments and suggestions aimed at improving the clarity, rigor, and completeness of the manuscript:

  • Abstract: It would be helpful to include more descriptive information about the sample (e.g., mean age, sex distribution) to provide a clearer overview of the study population.
  • Keywords: Please ensure that the keywords are listed in alphabetical order, in accordance with academic conventions.
  • Introduction: This section requires substantial revision. The variables under investigation (life satisfaction, psychological distress, and worries) should be more clearly defined and contextualised.
  • Introduction: the authors are encouraged to reference more longitudinal or trend studies on mental health during the COVID-19 pandemic to support or contrast their findings. The following examples are recommended, and the authors are invited to expand this list through a personal literature review:

Hyland et al., 2021; https://doi.org/10.1016/j.psychres.2021.113905

Gori & Topino, 2021; https://doi.org/10.3390/ijerph18115651

  • Methods: It is unclear whether the measures used in the study were validated instruments or ad hoc questionnaires. This should be clarified.
  • Methods: If validated instruments were used, the authors should cite the validated version in the participants' language.
  • Methods: If the instruments were developed ad hoc, it is important to include additional psychometric analyses (e.g., confirmatory factor analysis indices) to support their reliability and validity.
  • Results: This section is clearly written and well-structured. No major issues were identified.
  • Conclusions: The conclusion section should be significantly expanded. In particular, it should provide a more detailed discussion of the practical implications of the findings, including potential applications in public health, policy, or clinical settings.

Best regards,

Author Response

Reviewer: 4

Dear Review

Thank you for your comments. We have now changed after the reviews suggestions.

Daniel Lindberg

Thank you for the opportunity to review this study entitled “The Pandemic’s Impact on Mental Well-Being in Sweden. A Longitudinal Study on Life Satisfaction, Psychological Distress, and Worries” (ijerph-3630655).The study addresses the impact of the COVID-19 pandemic on mental health in Sweden, with a particular focus on the relationships among life satisfaction, psychological distress, and worries, as well as demographic and pandemic-related factors such as age, sex, education level, severe illness, and job loss. The sample includes 1,241 Swedish individuals. The manuscript deals with a highly relevant topic and overall represents a valuable contribution to the literature. One of its major strengths lies in its longitudinal design. However, several aspects of the paper would benefit from clarification and revision before it can be considered for publication. Below, I outline specific comments and suggestions aimed at improving the clarity, rigor, and completeness of the manuscript:

Abstract: It would be helpful to include more descriptive information about the sample (e.g., mean age, sex distribution) to provide a clearer overview of the study population.

Response: We agree and have added:

Line 20-21 among 588 men (mean age 54.9 years), and 653 women (mean age 52.9 years).

__________________________

Keywords: Please ensure that the keywords are listed in alphabetical order, in accordance with academic conventions.

Response: Thanks for your comment, we have listed the keywords in alphabetical order.

__________________________

Introduction: This section requires substantial revision. The variables under investigation (life satisfaction, psychological distress, and worries) should be more clearly defined and contextualised.

Response: We agree and have added:

Line 63-69: The COVID-19 pandemic has significantly impacted on the mental health of the population, leading to profound effects on individual and collective well-being both during and likely after the crisis [13]. An Irish study describes that the prevalence of major depression and generalized anxiety disorder did not increase because of, or during the early phase of the COVID-19 pandemic [14].

Line 85-92: The COVID-19 pandemic is described in Slovakia, having lasting negative effects on the mental and physical health of young adults, significantly impacting their overall quality of life [21]. Sweden welfare system with its universalistic policies, relatively generous and equalizing social and economic benefits in combination with its comparatively different approach to the Covid-19 pandemic focusing on public compliance with recommendations and personal responsibility makes it an interesting case for scientific enquiry [23].

__________________________

Introduction: the authors are encouraged to reference more longitudinal or trend studies on mental health during the COVID-19 pandemic to support or contrast their findings. The following examples are recommended, and the authors are invited to expand this list through a personal literature review:

Response: We agree and have added:

Line 63-69: The COVID-19 pandemic has significantly impacted on the mental health of the population, leading to profound effects on individual and collective well-being both during and likely after the crisis [13]. An Irish study describes that the prevalence of major depression and generalized anxiety disorder did not increase because of, or during the early phase of the COVID-19 pandemic [14].

Line 85-92: The COVID-19 pandemic is described in Slovakia, having lasting negative effects on the mental and physical health of young adults, significantly impacting their overall quality of life [21]. Sweden welfare system with its universalistic policies, relatively generous and equalizing social and economic benefits in combination with its comparatively different approach to the Covid-19 pandemic focusing on public compliance with recommendations and personal responsibility makes it an interesting case for scientific enquiry [23].

Line 425-431: However, these results need to be interpreted in relation to Sweden's approach to the pandemic, emphasizing voluntary guidelines over strict mandates, highlights an intriguing balance between personal responsibility and public health measures. This strategy contrasts sharply with the more stringent policies adopted by countries like Italy and France during the same period [33]. This may have mitigated severe distress and worries during the pandemic.

__________________________

Methods: It is unclear whether the measures used in the study were validated instruments or ad hoc questionnaires. This should be clarified. If validated instruments were used, the authors should cite the validated version in the participants' language. If the instruments were developed ad hoc, it is important to include additional psychometric analyses (e.g., confirmatory factor analysis indices) to support their reliability and validity.

Response: Thank you for the comment about additional psychometric analyses, but we sadly have not done this. Although, we have added:

Line 102-103: This panel has been widely used in research and featured in studies published in international journals [24,25].

Line 113-114: Prior studies using the same data have showed that the dropout only has minimal impact on the results during the COVID‐19 pandemic [25].

__________________________

Results: This section is clearly written and well-structured. No major issues were identified.

Conclusions: The conclusion section should be significantly expanded. In particular, it should provide a more detailed discussion of the practical implications of the findings, including potential applications in public health, policy, or clinical settings.

Response: We agree and have added:

Line 482—493: Our findings highlighted significant sex differences, with women reporting more negative outcomes in life satisfaction, psychological distress and worries, suggesting the need for gender-sensitive public health interventions. Recovery patterns showed that while short-term distress eased, long-term impacts on life satisfaction endured, emphasizing the importance of sustained mental health support. Younger adults (18–29) faced the greatest dissatisfaction, while older adults (65–79) exhibited remarkable resilience.

These insights underline the need for tailored policy responses sensitive to gender and age vulnerabilities, integrating psychosocial dimensions into broader strategies to promote recovery, resilience, and well-being across diverse demographic groups. 

Round 2

Reviewer 2 Report

Comments and Suggestions for Authors

I appreciate the authors' efforts to revise the manuscript and respond to the previous comments.
In my opinion, the manuscript has been sufficiently improved and can be accepted in its present form.

Author Response

Dear Editor in Chief,

Thank you for your comments. We have now changed after the reviews suggestions.

Daniel Lindberg

I do find you have responded very well to the suggestions of the reviewers. I have only a few minor suggestions to make:
1) It is still confusing that a higher score of life satisfaction means poorer (lower) life satisfaction. Would it not be possible to talk about "life dissatisfaction" instead (like you also talk about negative emotions like "distress" and "worries")

Answer: Thanks, we have changed life satisfaction to "life dissatisfaction".

2) Line 121: your figures are clearly no histograms, but maybe "line graphics". And these figures are not just descriptive but report analytical findings.

Answer: Thanks, we have changed to "line graphics".

3) Table 1 is still very long and goes over 2 pages. That makes reading more difficult. What if you change it into 3 tables (e.g. table 1a, 1b, and 1c)?

Answer: Thanks, we have changed to 3 tables (e.g. table 1a, 1b, and 1c).

4) in the footnote of table 1, you also write "3.2. Sex, age, and education differences", which is clearly not a footnote to the table but a new chapter headline. But you have the same headline also in line 169. I agree that the headline could/should come before figure 1.

Answer: Thanks, we have changed to the headline to before figure 1.

5) lines 162-164: it is simply not correct that you found significant sex differences with life satisfaction! Refer also to line 170! The whole new paragraph (lines 161-168) is just a summary of the findings reported from line 170 onward. Yes, you could summarize the results in the discussion chapter. But it is not necessary here!

Answer: Thanks, we agree and removed lines 161-168.